# Statistical Comparison and Assessment of Four Fire Emissions Inventories for 2013 and a Large Wildfire in the Western United States

Sam D. Faulstich [1,*], A. Grant Schissler [2], Matthew J. Strickland [3] and Heather A. Holmes [1]

1   Department of Chemical Engineering, University of Utah, Salt Lake City, UT 84112, USA; heather.holmes@chemeng.utah.edu
2   Department of Mathematics and Statistics, University of Nevada, Reno, NV 89557, USA; aschissler@unr.edu
3   School of Public Health, University of Nevada, Reno, NV 89557, USA; mstrickland@unr.edu
*   Correspondence: sam.faulstich@utah.edu

**Abstract:** Wildland fires produce smoke plumes that impact air quality and human health. To understand the effects of wildland fire smoke on humans, the amount and composition of the smoke plume must be quantified. Using a fire emissions inventory is one way to determine the emissions rate and composition of smoke plumes from individual fires. There are multiple fire emissions inventories, and each uses a different method to estimate emissions. This paper presents a comparison of four emissions inventories and their products: Fire INventory from NCAR (FINN version 1.5), Global Fire Emissions Database (GFED version 4s), Missoula Fire Labs Emissions Inventory (MFLEI (250 m) and MFLEI (10 km) products), and Wildland Fire Emissions Inventory System (WFEIS (MODIS) and WFEIS (MTBS) products). The outputs from these inventories are compared directly. Because there are no validation datasets for fire emissions, the outlying points from the Bayesian models developed for each inventory were compared with visible images and fire radiative power (FRP) data from satellite remote sensing. This comparison provides a framework to check fire emissions inventory data against additional data by providing a set of days to investigate closely. Results indicate that FINN and GFED likely underestimate emissions, while the MFLEI products likely overestimate emissions. No fire emissions inventory matched the temporal distribution of emissions from an external FRP dataset. A discussion of the differences impacting the emissions estimates from the four fire emissions inventories is provided, including a qualitative comparison of the methods and inputs used by each inventory and the associated strengths and limitations.

**Keywords:** fire; Bayesian statistics; air quality; wildfire smoke; Rim Fire





## 1. Introduction

Wildland fires harm humans and the environment. Impacts range from environmental destruction to serious health complications from smoke inhalation [1,2]. Wildland fire smoke inhalation can cause health complications, such as headaches and shortness of breath, and exacerbate existing conditions such as asthma and COPD [3–5]. Studies have also shown a link between wildland fire smoke exposure and non-fatal heart attacks [2,6]. $PM_{2.5}$ emissions from wildland fires may be more harmful than other sources of $PM_{2.5}$ due to the differences in chemical composition [7,8].

Atmospheric processes can transport smoke hundreds of miles, affecting communities far from the physical fire [9]. Wildland fire smoke transported through the atmosphere varies in concentration and composition after transport, impacting air quality and health downwind uniquely [2]. Understanding the downwind impact of wildland fire smoke requires knowing the concentration of the transported chemical constituents [10]. It is difficult to measure the characteristics of a smoke plume directly. Directly collected measurements lack generalizability due to the variable nature of fire behavior and emissions [11]. On the

ground, ambient air quality monitoring networks miss data related to the smoke plumes above the surface and collect information on all sources of pollution in the air (i.e., vehicle and industrial emissions). It is impossible to measure all the information needed to estimate emissions in the field, and once the atmosphere transports the smoke plume, the composition changes significantly [12]. Knowing the type and amount of pollution emitted by a wildland fire is challenging.

A fire emissions inventory is a mathematical representation of emissions that uses various inputs about the fire, land, and vegetation (biomass) burned to estimate the amount and type of pollution from an individual fire. Fire emissions inventories provide information about the smoke before it is transported and aged in the atmosphere and can be used as inputs to atmospheric dispersion models (e.g., HYSPLIT (HYbrid Single-Particle Lagrangian Integrated Trajectory model)) and chemical transport models (i.e., CMAQ (Community Multiscale Air Quality model)). These atmospheric models simulate the smoke plume concentration changes due to atmospheric transport. Both types of atmospheric models use mathematical representations of the physical processes to simulate atmospheric conditions. By combining fire emissions inventory estimates and atmospheric models, human exposure can be estimated in areas impacted by smoke plumes, which is crucial to understanding health outcomes.

Comparisons of fire emissions inventories have found variability between inventory results. In one study, fire area was the most significant driver of emissions variability, with fuel characterization and consumption determination methods also impacting emissions estimates [13]. Many of the differences between emissions inventories are most apparent at the smaller regional scale, as studies have reported less disagreement between inventories for larger country-wide and global scale studies compared to the smaller regional scale studies [14–17]. The variability and uncertainty of each input into the emission estimates makes it difficult to attribute the final variability in each emissions inventory to a specific data source; thus, determining the accuracy of a fire emissions inventory is challenging [16].

This paper provides a comparison of four fire emissions inventories and their associated products: Fire INventory from NCAR (FINN version 1), Global Fire Emissions Database (GFED version 4 s), Missoula Fire Labs Emissions Inventory (MFLEI (250 m) and MFLEI (10 km) products), and Wildland Fire Emissions Inventory System (WFEIS (MODIS) and WFEIS (MTBS) products). Results include a direct comparison of emissions estimates from each inventory for 2013 and the Yosemite Rim Fire. In addition to quantitatively comparing the estimates from each fire emissions inventory, a qualitative comparison using a Bayesian model is presented. Bayesian models determine influential points in the modeled distribution, where a point represents daily emissions amounts. These influential points help investigate how each emissions inventory represents the conditions for a specific day. We provide a discussion using the Bayesian model to identify the differences observed across emissions inventories. Results indicate that FINN and GFED likely underestimate emissions, while the MFLEI products likely overestimate emissions. No fire emissions inventory matched the temporal distribution of emissions from an external FRP dataset.

## 2. Fire Emissions Inventories and Satellite Remote Sensing

Fire emissions inventories estimate the amount and type of air pollution emissions from individual fires. The amount of area burned by the fire, the amount of each fuel type (biomass) in that burned area, and an emissions factor drive the emissions intensity from a fire [15]. Burned area is crucial to the amount of emissions released by a fire because the amount of area burned by the fire determines how much fuel was available for the fire. A fire that burns more area has access to more fuel and thus can create more emissions. Fuel characteristics also impact fire behavior, affecting fire emissions [18]. It is important to know the amount and type of biomass to determine fire characteristics and emissions. Because different biomass types release different emissions compounds in different amounts, an

emissions factor is crucial to determining emissions from a wildfire. The relationship between these variables is defined mathematically in Equation (1) [13].

$$\text{Amount of Species}_i \text{Emitted} = A * B * C * EF_i \tag{1}$$

*A* represents fire size, *B* represents available biomass, *C* represents combustion completeness, and $EF_i$ represents the emissions factor for the species of interest (*i*). The emissions factor links the amount of emissions released to dry matter burned. Fire emissions inventories rely heavily on remotely sensed data, often from satellites, to determine many of these input variables.

### 2.1. The Fire INventory from NCAR (FINN)

The Fire INventory from NCAR (FINN) [17] provides daily estimates globally at a 1 km spatial resolution. FINN version 1.5 is available for 2002 through 2020. FINN was primarily developed as a consistent emissions framework for atmospheric chemistry and air quality models. Therefore, FINN provides extensive information on chemical speciation, offering emissions estimates for over 40 different species.

The main advantage of FINN lies in its ability to produce data that is ready for input into chemical transport models in near real-time. FINN also uses rapidly available data sources, creating emissions estimates within a few hours of a satellite overpass. The MODIS burned area product is not available this quickly as the data processing for the burned area product can take months. Significant uncertainties in the assumed burned area product reduce the advantage of this rapidly available data when it is not being used for a near real-time simulation. FINN's use of the MODIS active fire product means that FINN can miss small fires because of remote sensing limitations. FINN also overestimates the size of small fires, in addition to misidentifying land cover and having significant uncertainties in fuel loadings and combustion completeness [17]. The generic land cover assignments used to apply emissions factors and fuel loading are broad, meaning that the average values of these variables applied to these regions do not always represent actual conditions [17]. FINN is in the process of updating its emissions inventory, and that update may address some of the disadvantages of this product.

### 2.2. Global Fire Emissions Database

The Global Fire Emissions Database (GFED) [16] is a monthly emissions inventory with a spatial resolution of 0.25° × 0.25° ( 27 km × 27 km). Additional data from satellite remote sensing instruments redistributes monthly emissions to daily and 3-hourly scales with diurnal cycles based on a method from Mu et al. [19]. GFED reports emissions in terms of dry matter. Dry matter must be converted to emissions using Equation (1) and the data provided by GFED. This method makes it easy to update the emissions factors used. GFED provides emissions factors for over 40 species and can also be used in chemical transport models. GFED version 4.1s provides an updated burned area product with an algorithm for small fire emissions. GFEDv4.1s is available from 1997–2020. This version of GFED includes updated fire severity estimates for boreal regions, fuel consumption for areas that frequently burn, and emissions factors for temperate and boreal areas.

GFED was developed primarily for global studies of climate and fire interactions, reflected by GFED's large spatial resolution [16]. According to regional studies, GFED is reliable for estimating emissions from large fires in specific locations, but has difficulties representing small fires and fires in certain regions [20,21]. Emissions estimates included in GFEDv4s are more reliable than estimates created by previous versions of GFED due to the numerous updates, but the remaining uncertainties are significant and challenging to quantify. The inclusion of small fires is a crucial step forward, but the limitations of remote sensing mean that the small fires product has significant uncertainties. An improvement on the original burned area dataset from MODIS [22], based on Randerson et al. [23], is used to determine a small fire burned area, but the dataset resolution is coarse compared to the size of the fires, introducing resampling error.

However, the main advantage of GFEDv4s is still the inclusion of small fires. Even though the small fire estimates have high uncertainty, GFED is the only fire emissions inventory with a specific algorithm to include small fires. The primary disadvantages of GFED are in the spatial and temporal resolution. GFED's large spatial resolution makes it less useful for regional-scale studies. The temporal profile used to rescale emissions to daily and 3-hourly scales may not appropriately represent the temporal nature of fire emissions, particularly in the western United States. For example, there is a significant nocturnal component to the emissions from fires in California [24,25]. The temporal profile used in GFED does not reflect this critical nocturnal component, meaning the rescaled data may inaccurately represent the temporal distribution of emissions.

### 2.3. Missoula Fire Lab Emissions Inventory (MFLEI)

The Missoula Fire Lab Emissions Inventory (MFLEI) [26] reports daily fire emissions from 2003–2015 for the contiguous United States. MFLEI gives estimates of the daily emissions of carbon monoxide, carbon dioxide, methane, and $PM_{2.5}$. MFLEI offers fire emissions at a 250 m × 250 m resolution (MFLEI (250 m)) and a spatially aggregated product at 10 km × 10 km spatial resolution (MFLEI (10 km)). The 10 km spatially aggregated product combines the data of the 250 m pixels into the larger spatial resolution using an uncertainty estimation approach [15], making the 10 km product better suited for atmospheric modeling applications. However, due to the limited number of species in MFLEI, the emissions estimates cannot directly be used in a chemical transport model.

The MFLEI method contains many updates that the developers promise increase accuracy in determining fire emissions, but this promised accuracy increase is difficult to quantify. The primary innovations in MFLEI come from updated emissions factors and a new wildland fuels map. The wildland fuels map forms the basis of fuel loading, which provides emissions estimates when combined with emissions factors [15]. Using specific emissions factors for different areas also reduces error compared to using a single emissions factor for the entire United States. Updating these inputs can significantly impact the emissions estimates from the fire emissions inventory. MFLEI still reports significant uncertainties in daily fuel consumption and $PM_{2.5}$ emissions. MFLEI also reports much larger $PM_{2.5}$ emissions in the west than other fire emissions inventories, likely related to the significantly higher emissions factor [15].

### 2.4. The Wildland Fire Emissions Inventory System (WFEIS)

The Wildland Fire Emissions Inventory System (WFEIS) [27] is a fire emissions inventory for the contiguous United States, with a daily temporal resolution and 1 km × 1 km spatial resolution. WFEIS allows for convenient retrieval and viewing of emissions estimates from their website, which also offers yearly fire emissions data at-a-glance. WFEIS provides emissions estimates for 2000–2020 and includes emissions for $CO_2$, CO, $CH_4$, $PM_{2.5}$, $PM_{10}$, and non-methane hydrocarbons.

WFEIS describes itself as a "system that provides open access to the modeling tools needed to quantify emissions from past fires" [27]. WFEIS developers do not consider their product an emissions inventory but an aggregation of tools that estimate input parameters for quantifying emissions. WFEIS has not developed methods of quantifying input variables and determining emissions. Instead, they use other tools to determine the input variables used to estimate emissions.

An advantage of the WFEIS method is that fuel consumption estimates are closer to measured values, especially the temporal variability, because they use a product to determine these values that is focused solely on fuel consumption [27]. WFEIS has higher greenhouse gas emissions and fuel consumption estimates than other inventories. There are several ways WFEIS approaches fuel consumption differently than other inventories (e.g., assigning smoldering and flaming consumption separately) that could cause an increase in WFEIS emissions estimates. Other advantages of WFEIS are the ease of data retrieval and use and the number of different burned area products available. However, the

ability to adjust the methods used to determine input variables is unavailable, as separate programs determine the input variables outside WFEIS. Another disadvantage is the high greenhouse gas emissions, as it is unknown which part of the WFEIS framework causes this increase.

*2.5. Satellite Remote Sensing*

The Visible Infrared Imaging Radiometer Suite (VIIRS) is an instrument onboard the polar-orbiting Suomi National Polar-orbiting Partnership (NPP) satellite. VIIRS products provide information on fire conditions to provide additional context for evaluating the fire emissions inventories. Multiple products based on remote sensing algorithms are available from the VIIRS instrument. VIIRS fire related products include fire radiative power (FRP), detection confidence level, and fire location. The remote sensing algorithm determines fire information based on thermal anomalies (i.e., detections of high-temperature pixels) [28]. These thermal anomalies are detected by an algorithm applied to the radiometer data, which includes information on the thermal infrared temperature of the pixels, allowing for the algorithm to detect differences in temperature between a pixel and those surrounding it. Fire radiative power is related to the fire size and the amount of emissions released. Because VIIRS is a satellite-based data set not used to create fire emissions inventories, it gives additional information on fire characteristics that can be used to understand the differences between fire emissions inventories.

**3. Methods**

A combination of techniques, from direct comparison to Bayesian statistical analysis, were used to compare the outputs of each fire emissions inventory. Because there is no evaluation dataset to determine the accuracy of each fire emissions inventory, using multiple types of comparisons helps uncover the variability in each emissions inventory. Each inventory's emissions estimates were compared for an annual case (2013) and a multi-day fire event in California (Yosemite Rim Fire). The geographic domain for 2013 included Washington, Oregon, California, Nevada, Arizona, Idaho, and part of Utah. A smaller geographic boundary based on the burned area information from CalFire was used for the Yosemite Rim Fire [29]. The period of the Yosemite Rim Fire was 17 August to 06 September 2013 [29]. The same geographic domains and times were used to filter the data from each emissions inventory. The spatial filter was exclusive, so it was excluded from the data if only part of a pixel was within the geographic boundary.

*3.1. Direct Comparison*

Each fire emissions inventory's estimates of burned area and CO, $CO_2$, $CH_4$, and $PM_{2.5}$ emissions were summed, both daily and annually, and compared for 2013 and the smaller case study of the Yosemite Rim Fire. While there are four inventories, there are six products to compare: FINNv1.5, GFEDv4s, MFLEI (250 m), MFLEI (10 km), WFEIS (MODIS), WFEIS (MTBS), shown in Table 1. The daily emissions were compared to understand how each inventory estimates the variability of fire activity day to day. Burned pixels were mapped using the latitude and longitude reported by the emissions inventory to investigate the spatial variation between each fire emissions inventory. The Yosemite Rim Fire provides insight into the differences in data reported by each inventory for a single, large wildfire. Understanding how fire emissions inventory estimates differ over the Yosemite Rim Fire case study is essential for understanding fire emissions inventory performance during the large wildfire events typical in the western United States, where existing fire behavior models may not capture fire behavior well [24,25]. An additional comparison was made between the fire emissions inventories and VIIRS FRP data. To have comparable results between all emissions inventories and VIIRS FRP, the emissions or FRP for each day were normalized to represent the percentage of annual emissions or FRP emitted on that day.

**Table 1.** Summary of fire emissions inventory characteristics: spatial resolution, number of data points for emissions estimates in 2013, the years of available data, temporal resolution, and the reference for the fire emissions inventory are included.

| | FINN | GFED | MFLEI (250 m) | MFLEI (10 km) | WFEIS (MODIS) | WFEIS (MTBS) |
|---|---|---|---|---|---|---|
| Spatial Resolution | 1 km$^2$ | 770 km$^2$ | 0.063 km$^2$ | 100 km$^2$ | 1 km$^2$ | 0.0009 km$^2$ |
| Data Points | 7985 | 921 | 119,669 | 3430 | 9272 | 159 |
| Years Available | 2002–2020 | 1997–2021 | 2003–2015 | 2003–2015 | 2000–2021 | 1984–2019 |
| Temporal Resolution | Daily | Monthly, Daily, 3-hourly | Daily | Daily | Daily | Fire Start Date |
| Reference Paper | Wiedinmyer et al. [17] | van der Werf et al. [16] | Urbanski et al. [15] | Urbanski et al. [15] | French et al. [27] | French et al. [27] |

*3.2. Bayesian Statistical Analysis*

The daily PM$_{2.5}$ estimates from each fire emissions inventory were used to create Bayesian models that further investigate the daily emissions distributions provided by each fire emissions inventory. Multiple Bayesian models were created for each fire emissions inventory, fit to actual inventory data. Understanding these Bayesian model parameters allows us to infer information about the distribution of emissions estimates from each fire emissions inventory. These models can provide an innovative methodology to detect each point's influence on the Bayesian model distribution. These influence metrics quantify how well the Bayesian model distribution predicts specific time points. Points that outlie and influence the overall distribution can then be investigated using external data sources (i.e., the VIIRS remote sensing products).

The Bayesian model created for each fire emissions inventory relates the day of the year and the amount of PM$_{2.5}$ emissions reported by each fire emissions inventory. These models were created for 2013, the 2013 fire season (June–September), and the Yosemite Rim Fire. When creating a time series, it is essential to check for autocorrelation and correct it. An autoregressive (AR) model was also created for each fire emissions inventory to check and correct for autocorrelation. Correcting autocorrelation is crucial because many statistical analyses require that the data be independent of each other, and if they are correlated in time, they are not independent. The input data is formulaically corrected if autocorrelation is present. An AR model is advantageous in this context because the Bayesian model now represents how the previous days' fire characteristics relate to the following days once corrected. It is more likely that the amount of emissions released is related to the amount of emissions released the day before than to what day of the year it is. An autoregressive Bayesian model was created for each fire emissions inventory to determine how PM$_{2.5}$ emissions reported for each day are related to the next day's emissions. In all cases, the autoregressive model was a better fit than the non-autoregressive model. While the Bayesian models for each fire emissions inventory are not directly comparable to the Bayesian models of the other fire emissions inventories, each model can provide other directly comparable information, such as cross-validation statistics.

Now, we mathematically describe our AR model (Equation (2)). First, we denote the daily PM$_{2.5}$ emissions in grams as the outcome variable $Y$. The model assumes that $Y$ is normally distributed with its mean conditional on the previous day's PM$_{2.5}$. Since we only explicitly include the previous day, a single lag, this specific AR is denoted AR(1). Note, this structure induces longer-range autocorrelation as a consequence of this correction. To complete the Bayesian model, prior distributions are specified on each parameter outcome of the model. We choose to employ weakly informative priors to improve estimation efficiency and avoid entirely unreasonable parameter values. At the same time, we allow the posterior distribution to be predominantly influenced by the likelihood of the parameters conditional on the data. In the AR(1), we set weakly informative priors by specifying the constant values for $\alpha$ and $\beta$'s prior distributions to be skeptical, centered at 0, with

large variances (Equation (2)). The first line in Equation (2) below denotes an AR(1) model related to PM$_{2.5}$ emissions in grams for current day $n$, and the following three lines specify the prior distribution for the Bayesian model parameters.

$$
\begin{aligned}
Y_n &\sim normal(\alpha + \beta y_{n-1}, \sigma) \\
\alpha &\sim N(0, 10) \\
\beta &\sim N(0, 2.5) \\
\sigma &\sim exponential(rate = 1)
\end{aligned}
\tag{2}
$$

Finally, we select a prior for the exponential scale parameter with a rate of 1, the maximum entropy distribution for a positively constrained random variable, thereby improving predictive performance [30].

To estimate the posterior and predictive distributions, the R program RStan was used [30–32]. Markov Chain Monte Carlo (MCMC) was used to investigate the Bayesian model distributions. MCMC creates samples from a continuous random variable, with the probability density proportional to a known function (i.e., Equation (2)). The MCMC model was set up to obtain four chains of 2000 samples each from every model created. There must be enough samples for the chains to converge for the model to run well. When the chains converge, the model has run enough simulations to effectively create a reasonable posterior predictive distribution. The chain convergence for these models was investigated using ShinyStan, a part of the RStan package, and showed that all chains for each model converged.

### 3.3. Influential Points Investigation

An innovative way to use our statistical models to investigate fire emissions inventories is to use cross-validation techniques to determine influential points in the Bayesian model. The "leave one out" (loo) technique is used for cross-validation. This technique removes a single data point from the Bayesian model distribution and compares the new results to the results of the Bayesian model distribution when all points are included. The loo process in R provides an estimation of this refitting. This estimation uses the impact of each point on the posterior distribution, referred to as importance. Importance can be estimated without refitting the model. Unlikely observations will have more importance than expected observations, so investigating the influential points determines which points do not match the rest of the predicted model distribution and thus have the most influence on the overall distribution of the Bayesian model [30]. It is impossible to check every data point in a fire emissions inventory against daily fire reports and satellite images. Narrowing the investigation down to only influential points makes it possible to use other data to help evaluate if the fire emissions inventory reasonably captures the fire's characteristics for that specific point.

The influential points investigation uses Pareto Smoothed Importance Sampling (PSIS) plots created using leave one out cross-validation on the autoregressive models created for each fire emissions inventory. PSIS uses importance sampling to determine cross-validation, reporting information on the relative importance of each point in the Bayesian model distribution [33]. Outlying influential points are defined as points with a Pareto-k influence score outside of a specified threshold (k > 0.7). Above this threshold, error estimates from the PSIS sampling become unreliable, and thus, the PSIS model cannot accurately represent these points, meaning the point is outlying the predicted distribution of the Bayesian model [34,35]. Some models do not have influential points that outlie the Bayesian model distribution. In this case, it can still be helpful to investigate the most influential point of a Bayesian model distribution, even if it is not an outlier. This influential points investigation will not provide definitive answers on which fire emissions inventory best models real-world conditions. However, it provides additional context outside of the quantitative emissions estimates comparison on how the emissions inventories represent specific days.

## 4. Results

### 4.1. Spatial Distribution

The map of all products over the spatial domain for 2013 (Figure 1) shows that all the fire emissions inventories have a similar spatial distribution. GFED (Figure 1c) has few points due to the low spatial resolution. WFEIS (MTBS) has very few points because of how the burned pixels are assigned with the MTBS burned area product. The location reported by WFEIS (MTBS) (Figure 1e) is in the center of the fire area. MTBS does not provide the spatial extent of the burned area unless the burned area perimeter is incorporated, which it is not in WFEIS (MTBS). Each fire emissions inventory has burned pixels in similar areas, but some fire emissions inventories (i.e., MFLEI (250 m) and WFEIS (MODIS)) have more pixels in those fire locations. In the central part of Nevada, there are fires sensed by some emissions inventories that are not reported by other inventories.

The difference in spatial distribution between WFEIS (MODIS) and WFEIS (MTBS) is pronounced, though they are products from the same fire emissions inventory. WFEIS (MTBS) has fewer fire locations points, but there are WFEIS (MTBS) points where WFEIS (MODIS) does not have any points, such as in northwestern Arizona or the Oregon-Idaho border. MTBS does not assign temporal progression of burned area, so there are fewer fire pixels for the WFEIS (MTBS) product. MTBS also has a higher spatial resolution than MODIS, which explains why burned pixels with WFEIS (MTBS) are not included in the WFEIS (MODIS) product. The MFLEI products have a similar temporal resolution because the MFLEI (10 km) burned area is determined by aggregating the MFLEI (250 m) burned pixels (Figure 1d).

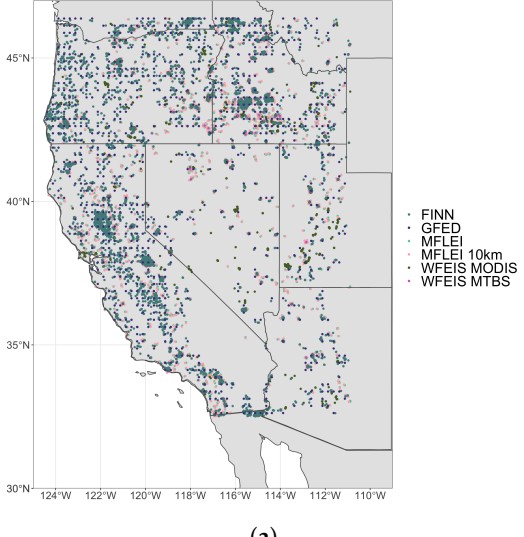

(a)

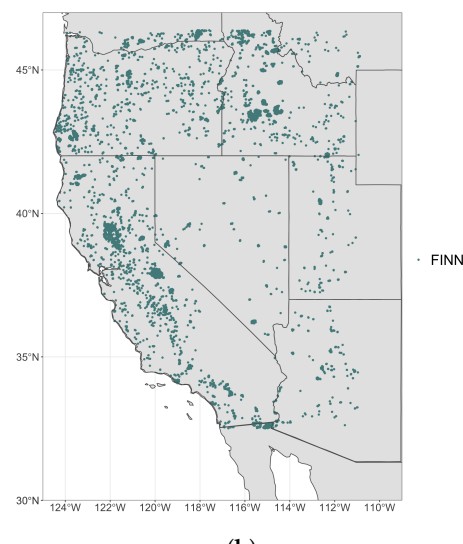

(b)

**Figure 1.** *Cont.*

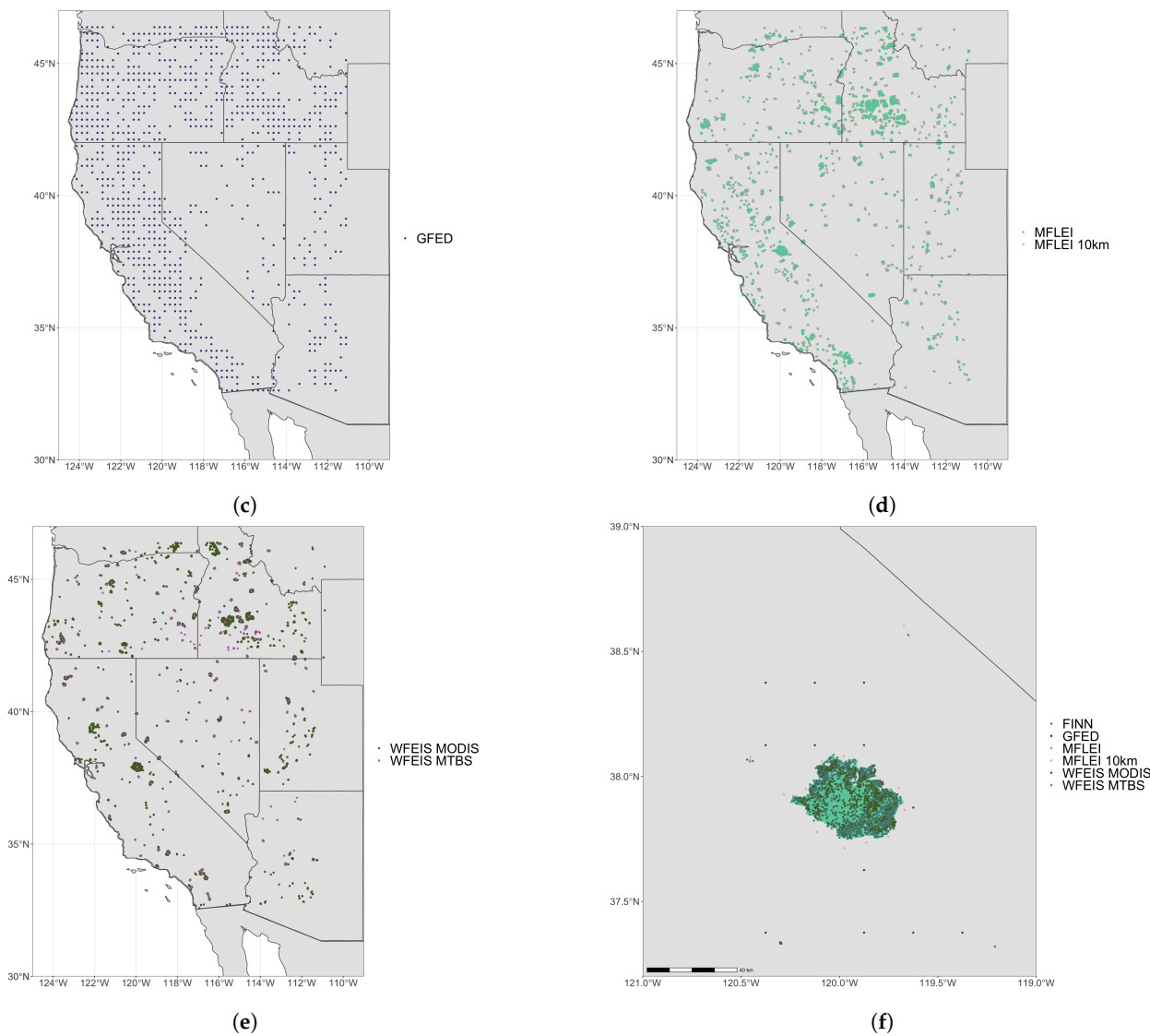

**Figure 1.** (**a**) All Inventories (**b**) FINN (**c**) GFED (**d**) MFLEI (250 m) + MFLEI (10 km) (**e**) WFEIS (MODIS) + WFEIS (MTBS) (**f**) Yosemite Rim Fire. Maps showing the burned area pixels reported by each fire emissions inventory. These maps show the difference in fire location between each inventory due to inventory resolution and methods.

*4.2. Annual Evaluation (2013)*

4.2.1. Burned Area

For the spatial domain, WFEIS (MTBS) had the largest burned area for 2013, followed by the MFLEI products, GFED, WFEIS (MODIS), and FINN, in that order (Figure 2). Each emissions inventory has a similar burned area for the entire year, with FINN reporting 84.5% of the burned area reported by WFEIS (MODIS). Most fire emissions inventories use MODIS data in their burned area products, explaining the similarity. WFEIS (MTBS) has the highest burned area due to the higher spatial resolution of the MTBS burned area product compared to the MODIS burned area product. FINN's assumed burned area method seems to underestimate emissions compared to the other inventories. All fire emissions inventories have a burned area maximum in August, but the spatial distribution of burned area in months with less fire activity varies greatly (Figure 3).

4.2.2. Emissions

All inventories show a very different total amount of emissions in 2013. MFLEI (250 m), WFEIS (MODIS), and WFEIS (MTBS) are similar in all emissions species. GFED and MFLEI

(10 km) are consistently lower than all other emissions inventories, with GFED the lowest (Figure 2). Each emissions inventory shows a similar temporal variability over the year, with prominent peaks in August (Figure 3). Though they all show a similar temporal variability, the daily emission magnitudes reported daily by each emissions inventory are vastly different. The magnitude of emissions from MFLEI (250 m) is much higher than all other inventories. FINN and GFED are well correlated in magnitude during the emissions peak of the year, but the temporal variation of GFED for the rest of the year is enormously different from what any other emissions inventory shows. GFED captures fires during the peak of fire season but does not detect fires during other parts of the year when fires are typically smaller. Each emissions inventory captures emissions maxima throughout the year similarly but reports different variability for periods with lower emissions.

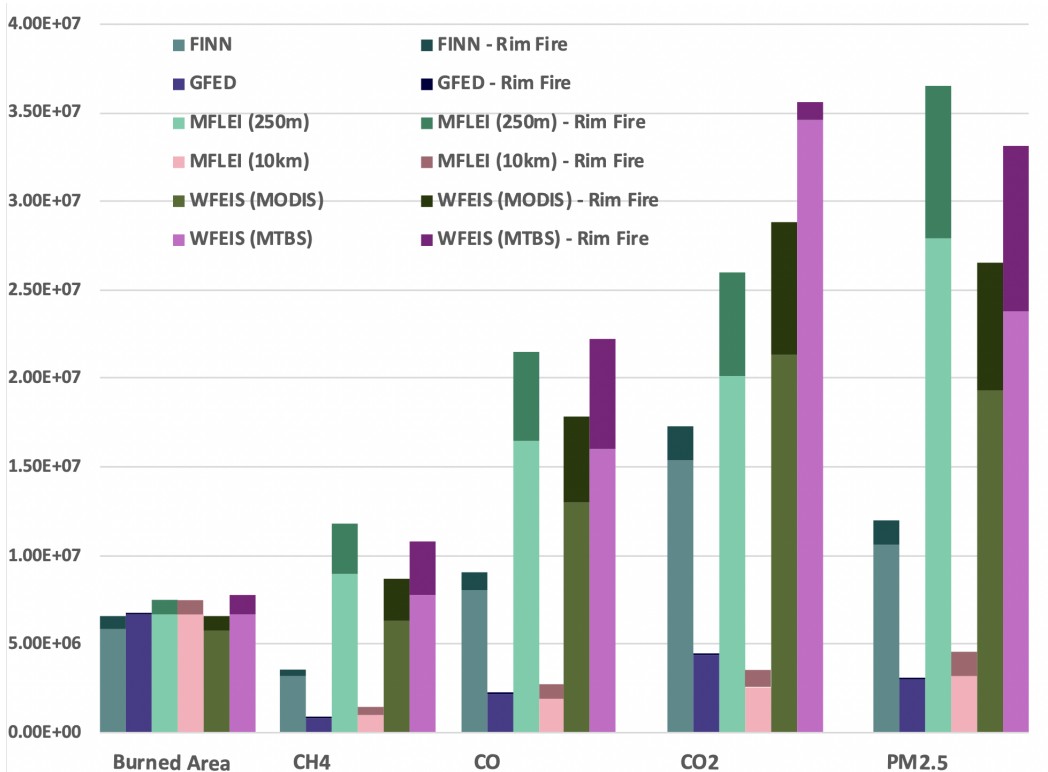

**Figure 2.** Burned area and emissions from each emissions inventory for 2013, with the top, darker color indicating the Yosemite Rim Fire emissions. For GFED, the Yosemite Rim Fire emissions are so much lower than the yearly emissions that the darker color cannot be discerned. Because each inventory reports drastically different emissions estimates, each category needed to be scaled differently to represent them all on the same graph. Burned area is reported in $m^2$, while all chemical species are reported in grams. Burned area is scaled by $10^{-3}$, $CH_4$ is scaled by $10^{-4}$, CO is scaled by $10^{-5}$, $CO_2$ is scaled by $10^{-6}$, and $PM_{2.5}$ is scaled by $10^{-5}$. The y-axis is represented linearly with a break at zero.

*4.3. Yosemite Rim Fire*

4.3.1. Burned Area

The burned area reported for the Yosemite Rim Fire was more variable than the yearly burned area (Figure 2). WFEIS (MTBS) has the highest burned area for the Yosemite Rim Fire, followed by WFEIS (MODIS), the MFLEI products, FINN, and GFED, respectively. The difference between GFED and FINN is approximately a factor of 10. The burned area reported by GFED for the Yosemite Rim Fire is 11.3% of the burned area reported by WFEIS (MTBS). Though MFLEI (250 m) has the most points, it does not have the highest burned area. Because burned area is a fundamental part of the emissions equation (Equation (1)), differences in burned area significantly impact emissions estimates. The large differences

between fire emissions inventories seen for a single fire event compared to the whole year show how inventory selection can impact work done using single-fire data from fire emissions inventories.

### 4.3.2. Emissions

Though the burned area is different, the variability between emissions for the Yosemite Rim Fire is similar to the 2013 total emissions. The WFEIS products and MFLEI (250 m) report similar emissions. GFED has the lowest emissions of all constituents, which is expected with the smallest burned area (Figure 2). Each emissions inventory shows a different magnitude of $PM_{2.5}$ emissions and temporal distribution of those emissions for the Yosemite Rim Fire (Figure 3). MFLEI (250 m), MFLEI (10 km), and WFEIS (MODIS) show similar temporal variation, which is likely because they all use the MODIS burned area product. GFED shows a distinctly different emissions profile that is difficult to discern due to the low emissions from GFED. FINN also shows a different temporal variation, with emissions increasing and decreasing from day to day in a sawtooth pattern. This pattern could be due to the assumed burned area product, with difficulties in satellite remote sensing (e.g., cloud cover) leading to instability in the number of thermal anomalies sensed each day. The MFLEI products and WFEIS (MODIS) show maximum emissions on the same day (22 August). FINN and GFED show maximum emissions on 26 August, but the other inventories also show an increase in emissions on that day.

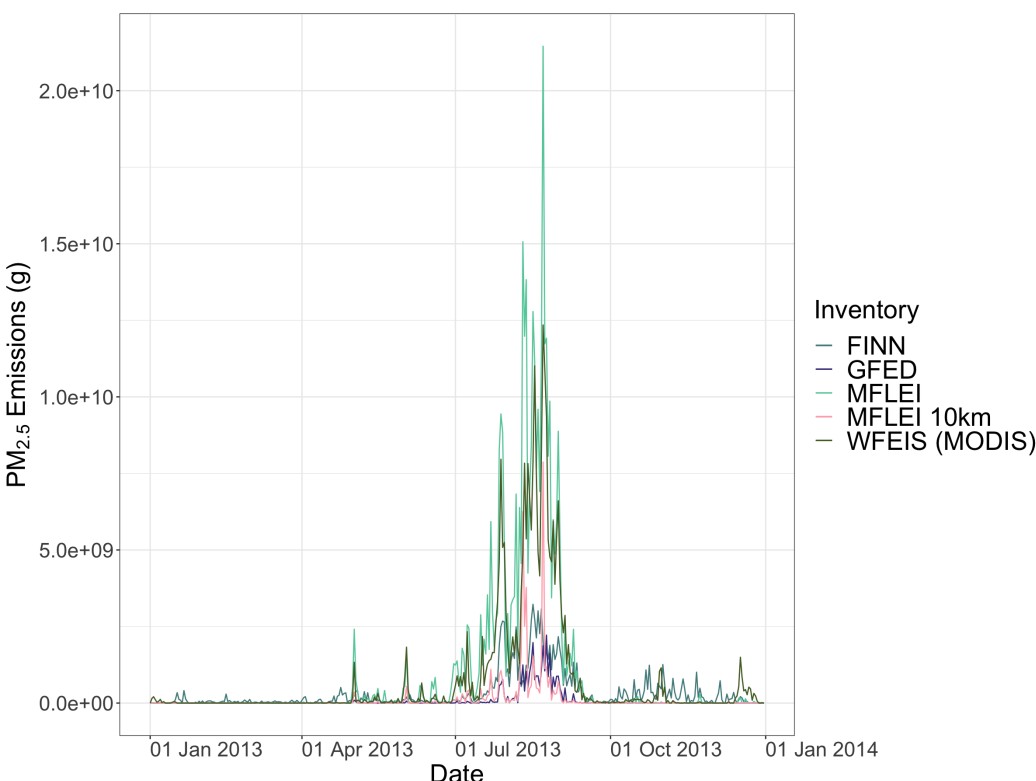

**Figure 3.** Daily $PM_{2.5}$ emissions in grams for each fire emissions inventory over 2013. Note: The y-axis is scaled linearly, with a break at the origin. All inventories show an increase in $PM_{2.5}$ emissions between July and October, but the magnitude of the increase varies significantly between inventories. There is also significant disagreement between inventories during lower fire activity months.

### 4.4. VIIRS Fire Radiative Power Comparison

In addition to providing data for investigating influential points, the VIIRS FRP can be compared with $PM_{2.5}$ estimates from each inventory. Using the time series of VIIRS FRP as an independent evaluation dataset, the temporal pattern in the emissions inventories can be investigated. The highest individual FRP values are in August (Figure 4). Interestingly,

there are also some high FRP values in July. This July increase in the FRP is not reflected in the fire emissions inventories temporal distribution of fire activity. This increase may be due to a hotter burning fire, which may not be well represented in fire emissions inventories as emissions inventories calculate emissions based solely on burned area. Because emissions are related to fire temperature, a hotter burning fire can have more emissions than a lower temperature fire of the same area.

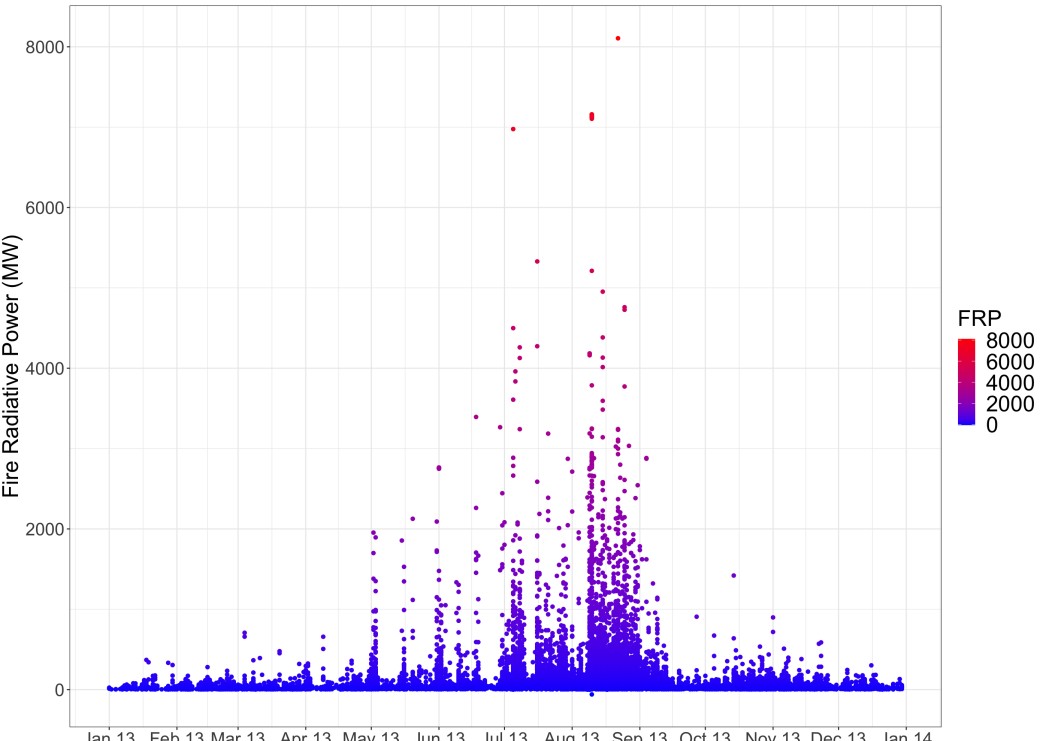

**Figure 4.** Fire radiative power (FRP) in megawatts from VIIRS. Each point represents the FRP of a single thermal anomaly detected by VIIRS. FRP is related to fire intensity and emissions amount. The VIIRS FRP is not included in any fire emissions inventories and provides an independent dataset for evaluation.

The data of the percentage of annual FRP emitted per day (Figure 5) shows that VIIRS captured the most fire activity on 16 August (6.87% of annual FRP), followed by 22 August (6.55%) and 10 August (5.31%). Only FINN captures the same maxima in fire activity as VIIRS, showing that 2.70% of yearly $PM_{2.5}$ emissions were emitted on 16 August, which is the maximum for FINN. The MFLEI products and WFEIS (MODIS) report their maximum annual percentage of $PM_{2.5}$ emissions emitted in a single day on 22 August, which is the second-highest day for VIIRS. GFED reports the highest percentage of annual $PM_{2.5}$ emissions emitted in a single day on 24 August, which is not in the top five highest days of the VIIRS data.

The annual time series of FRP can be compared with the time series from each inventory. Pearson correlation coefficients between the daily $PM_{2.5}$ emissions and the daily VIIRS FRP are shown in Table 2. MFLEI (250 m) has the highest correlation coefficient, followed by GFED. Based on the results shown above, these emissions inventories have not accurately represented physical conditions, so the strong correlation with VIIRS FRP is interesting. GFED captured the peaks in fire activity on 10 August, 16 August, and 22 August quite similarly to VIIRS FRP, but the GFED data aside from these days is not particularly close with the VIIRS FRP data. Because GFED captures the same maxima as VIIRS, the correlation increases, though the other data is not similar. The high correlation coefficient could also mean that MODIS satellite data used in the GFED disaggregation to the daily

time is similar to the VIIRS data, which is likely. MFLEI (250 m) also closely captures the fire activity peaks represented by VIIRS on 10 and 22 August and captures the 16 August peak, but less closely than GFED did. No other fire emissions inventories are close to the VIIRS data on 10 August or 16 August, thus leading to weaker correlations because they are not similarly capturing the three fire activity maxima seen by VIIRS in August.

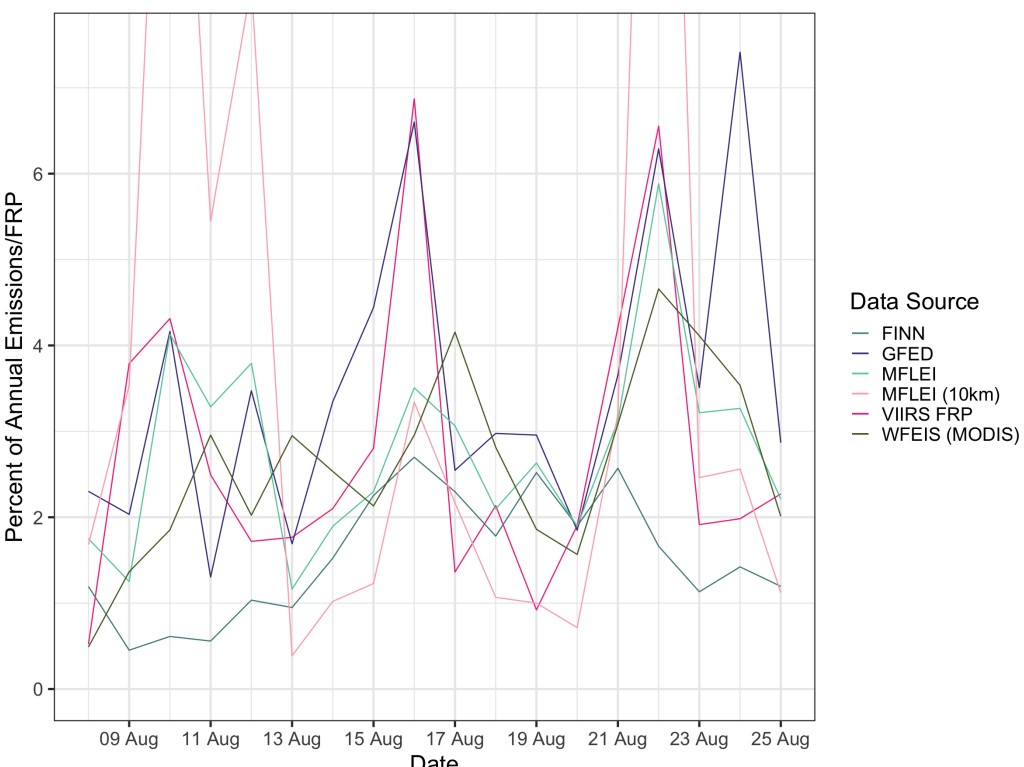

**Figure 5.** The percentage of annual FRP from VIIRS emitted per day with the percentage of annual $PM_{2.5}$ emitted per day from all fire emissions inventories for 9–25 August 2013. This period contains all of the frequently occurring influential points. The MFLEI (10 km) points that exceed the y-axis correspond to 13.57%, 8.19%, and 17.07%, in chronological order. VIIRS FRP shows peaks in fire activity on 10 August, 16 August, and 22 August. GFED captures these maxima closest, explaining the strong correlation between VIIRS and GFED. However, GFED does not closely agree with the VIIRS data aside from those three days.

*4.5. Bayesian Statistical Modeling*

4.5.1. Influential Points Investigation

Table 3 provides information on the influential points of each model for each fire emissions inventory and each temporal filter (annual, fire season, and Rim Fire). Fire season is defined as June through September. Modeling daily $PM_{2.5}$ emissions for only the fire season creates a temporal filter that removes many of the low or no emissions days that can be difficult for the Bayesian model to represent accurately. The influential points found in each inventory were compared with visual satellite images to evaluate fire activity on that date. These images are not included here because this inspection was done manually using NASA Worldview [36]. This investigation is an initial qualitative check to determine if fires are occurring in the spatial domain on the days of the influential points. All fire emissions inventories were found to pass the qualitative visual check using NASA Worldview, meaning that there were thermal anomalies detected in the spatial boundary on the days of the influential points. VIIRS FRP data (Figure 4) provides context for the fire activity of the surrounding days to provide quantitative analysis. Analyzing the fire activity captured by a fire emissions inventory for an influential point and relating it to the

fire activity captured by VIIRS helps explain why that point may have been outlying in the Bayesian model.

**Table 2.** Pearson correlation coefficients between daily, summed VIIRS FRP and daily, summed PM$_{2.5}$ for each emissions inventory. The correlation between VIIRS FRP, an external dataset, and daily PM$_{2.5}$ for each inventory assesses the trends captured by each emissions inventory compared to VIIRS FRP data. All correlation coefficients are statistically significant ($p \leq 0.05$).

| Inventory | Correlation |
|:---:|:---:|
| FINN | 0.66 |
| GFED | 0.84 |
| MFLEI (250 m) | 0.86 |
| MFLEI (10 km) | 0.77 |
| WFEIS (MODIS) | 0.80 |
| WFEIS (MTBS) | 0.20 |

FINN did not have an outlying point for the fire season Bayesian model, but the most influential point of the model occurred on 07 August. FINN reported significant PM$_{2.5}$ emissions for that day. This point was also influential in the 2013 Bayesian model for FINN. VIIRS reports several fires but a relatively low level of FRP. 07 August has a lower FRP than the days around it. FINN replicates the pattern seen in the FRP data, with 07 August having lower emissions than the days around it. The lower emissions for a single day may create a pattern that strongly influences the distribution of the Bayesian model, primarily since it occurs at a low point. The model likely predicts a smooth transition between 06 August and 08 August, causing this increase/decrease/increase pattern to outlie the predicted distribution of the model. For the Yosemite Rim Fire Bayesian model, FINN had no outlying influential points. The most influential point of the Yosemite Rim Fire Bayesian model was 28 August. FINN reports a large amount of PM$_{2.5}$ and burned area for this day. VIIRS FRP shows that 28 August has several fires but fewer high FRP points than the surrounding days. FINN does not represent the same trend as the VIIRS FRP on the surrounding days. Where VIIRS reports a decline in FRP from 26 August–28 August, FINN reports 27 August has lower emissions than the days around it. This is the same trend that was flagged in the previous outlying point. These single-day decreases may be causing these points to be marked as outlying the predicted Bayesian model distribution. Incorporating the VIIRS FRP data shows that FINN captures a different fire activity trend than VIIRS.

The most influential point of the fire season Bayesian model for GFED was 24 August. 24 August was also an outlying influential point for the 2013 Bayesian model. GFED showed a significant spike in emissions on 24 August and estimated the largest percentage of yearly PM$_{2.5}$ emissions on this day. The emissions of the days surrounding 24 August are also lower. Not only is the percentage of yearly emissions emitted on this day at a maximum, but the emissions of the days around it are also lower, and this pattern is likely why this point was selected as influential. A maximum that is different from its neighbors will impact the distribution. VIIRS FRP data for 24 August does not show the same increase as reported by GFED.

The influential point for GFED during the Yosemite Rim Fire occurred on 26 August. The Pareto-k influence value of this point is significant, greater than 1.5. This point is highly influential and outlying in the predicted distribution of the Bayesian model. Referencing the daily PM$_{2.5}$ graph (Figure 3), GFED shows an increase on 26 August, with the rest of the Yosemite Rim Fire emissions being low. It makes sense that this would be the most influential point of the model for GFED, as this is a point with some of the highest emissions in the entire model. The satellite imagery shows the Yosemite Rim Fire emits a large smoke plume on this day, so emissions are likely to be high. This reveals a limitation of the investigation of the influential points. From this single influential point, all that can be determined is whether GFED is representing this day reasonably or not. It does not provide any information on how emissions should progress from day to day. The remote sensing

imagery shows that the Yosemite Rim Fire had large smoke plumes for several days, so it is not reasonable to represent the temporal distribution of the Yosemite Rim Fire by having only one day of high emissions. While the investigation of the influential point cannot reveal any information about the temporal variability of the Yosemite Rim Fire, it did bring attention to this point, encouraging investigation of why this point influences the model so heavily. The investigation of this influential point revealed that GFED is not representing the temporal distribution of the Yosemite Rim Fire in a way that is close to reality. This is confirmed by comparing the percentage of annual $PM_{2.5}$ emissions emitted each day by GFED with the percentage of annual FRP for each day from VIIRS. Figure 5 shows that GFED does capture a few large spikes in fire activity similarly to VIIRS FRP, but GFED shows jagged, sawtooth points, where VIIRS FRP shows more consistent increases and decreases in emissions.

**Table 3.** Dates corresponding to the influential points for each fire emissions inventory Bayesian model (2013, fire season, and Rim Fire) and the number of fire pixels, $PM_{2.5}$ emissions (grams), and burned area ($m^2$) for each point. The percentage of annual emissions emitted on each day is also provided. An asterisk denotes the maximum percentage of annual emissions emitted. The Pareto-k influence value is only presented for points that were not outliers. Note: GFED does not provide pixel information for each fire.

| | Date | Model | Pareto-k Influence Value | Pixels | $PM_{2.5}$ (Grams) | Percent of Annual $PM_{2.5}$ Emissions | Burned Area ($m^2$) |
|---|---|---|---|---|---|---|---|
| FINN | 07 August | 2013, Fire Season | 0.29 | 34 | $7.50 \times 10^8$ | 0.62% | $3.04 \times 10^7$ |
| | 28 August | Rim Fire | 0.31 | 63 | $1.44 \times 10^9$ | 1.60% | $6.14 \times 10^7$ |
| GFED | 17 August | 2013 | – | – | $7.62 \times 10^8$ | 2.55% | $1.10 \times 10^8$ |
| | 24 August | 2013, Fire Season | – | – | $2.21 \times 10^9$ | 7.14% * | $1.47 \times 10^8$ |
| | 25 August | 2013 | – | – | $8.58 \times 10^8$ | 2.87% | $9.45 \times 10^7$ |
| | 27 August | 2013 | – | – | $2.39 \times 10^8$ | 0.80% | $4.77 \times 10^7$ |
| | 26 August | Rim Fire | – | – | $2.09 \times 10^8$ | 5.45% | $1.12 \times 10^8$ |
| MFLEI (250 m) | 10 August | 2013 | – | 8959 | $1.51 \times 10^{10}$ | 4.13% | $5.60 \times 10^8$ |
| | 22 August | 2013, Fire Season, Rim Fire | – | 3278 | $2.15 \times 10^{10}$ | 5.88% * | $1.19 \times 10^8$ |
| | 23 August | 2013 | – | 1904 | $1.17 \times 10^{10}$ | 3.22% | $1.19 \times 10^8$ |
| MFLEI (10 km) | 10 August | 2013, Fire Season | – | 48 | $2.51 \times 10^9$ | 13.57% | $5.60 \times 10^8$ |
| | 22 August | 2013, Fire Season, Rim Fire | – | 52 | $7.87 \times 10^9$ | 17.07% * | $2.33 \times 10^8$ |
| | 23 August | 2013, Fire Season, Rim Fire | – | 50 | $1.23 \times 10^9$ | 2.46% | $1.19 \times 10^8$ |
| WFEIS (MODIS) | 22 August | 2013, Rim Fire | 0.58 | 195 | $1.23 \times 10^{10}$ | 4.66% * | $2.16 \times 10^8$ |
| | 18 August | Fire Season | 0.39 | 176 | $7.46 \times 10^9$ | 2.81% | $1.24 \times 10^8$ |

MFLEI (250 m) had one outlier influential point for the fire season, 22 August. Again, this point was also influential in the 2013 model. This day had the highest percentage of the yearly emissions emitted, and closely matches the VIIRS percentage of yearly FRP emitted, so it is appropriate to have a spike here. Additionally, the days surrounding 22 August report lower emissions, pointing again to a pattern that will impact the distribution of a Bayesian model based on this data.

MFLEI (10 km) had three outlier influential points for the fire season, 10 August, 22 August, and 24 August, also outlying in the 2013 Bayesian model. 22 August is also the highest percentage of yearly emissions emitted in a single day for MFLEI (10 km), but the percentage is much higher than found in MFLEI (250 m) (i.e., 17% vs. 6%). 10 August reports the second-highest percentage of yearly emissions emitted in a single day for MFLEI (10 km). VIIRS FRP shows a peak in FRP on 10 August, but it is the third-largest peak for the VIIRS FRP. In MFLEI (10 km), 24 August shows a slight increase in emissions from 23 August, but because 23 August is so much lower than 22 August, an increase after this steep decrease can influence and outlie the distribution of the Bayesian model since it would likely predict a continued decrease.

MFLEI (250 m) had fewer influential points using the fire season Bayesian model than the annual model, but MFLEI (10 km) had the same influential points using the fire season and annual Bayesian models. For the Yosemite Rim Fire model, MFLEI (250 m) had an outlying influential point on 22 August. MFLEI (10 km) has outlying influential points on 22 and 23 August. These points were also found in the annual and fire season Bayesian models. This points to the differences that spatial aggregation can create, even when using the same data and methods.

WFEIS (MODIS) did not have an outlying influential point for the fire season Bayesian model. The most influential point occurred on 18 August. 18 August was not the maximum percentage of emissions, nor is it higher or lower than both days surrounding it. For this to be the most influential point of the model is particularly interesting. It may point to the influential points being "decision points" of a model or places where the algorithm must decide if it will increase or decrease at this point. WFEIS (MODIS) had no outlying influential points in the Yosemite Rim Fire Bayesian model. The most influential point was on 22 August. WFEIS (MODIS) reports significant $PM_{2.5}$ emissions for this day. 22 August had the highest percentage of yearly $PM_{2.5}$ emissions recorded by WFEIS (MODIS). As with other emissions inventories, this maxima makes sense as an influential point because it drives the distribution. Compared to the VIIRS data, WFEIS (MODIS) captures the peak in fire activity on 22 August but does not closely match VIIRS on other features of temporal fire activity on the days before and after the influential point.

The qualitative investigation of fire emissions inventories using satellite imagery from NASA Worldview proved that fires occurred in the spatial domain on all days of influential points. The VIIRS FRP dataset provided a comparison dataset for quantitative investigation of the influential points via an additional temporal record of fire activity. Many influential points were days with a high percentage of yearly emissions emitted on a single day. Comparing the estimations of the fire emissions inventories with the VIIRS FRP data for August, when all of the influential points took place, the fire emissions inventories capture some of the same maxima, with all inventories but FINN representing a large amount of fire activity on 22 August. There are not many similarities beyond this, and none of the fire emissions inventories closely represent the temporal profile of VIIRS or the other fire emissions inventories. Because the data has been normalized to represent the percent of annual emissions or FRP on that day, this investigation cannot provide information about if a fire emissions inventory is correctly estimating the amount of emissions. It can only provide information on if the inventories and VIIRS are representing the same temporal fire activity profile, which they are not.

### 4.5.2. Frequent Influential Points

Comparing the emissions reported by each fire emissions inventory for the most often occurring influential points provides insight into how each emissions inventory represents emissions for days frequently reported as influential. In this investigation, the frequently reported influential points are 10 August (3 occurrences), 22 August (8 occurrences), and 23 August (4 occurrences). MFLEI (250 m) reports the highest emissions for all frequent influential points. WFEIS (MODIS) reports the subsequent highest emissions for all frequent influential points (Table 4). FINN reports the lowest emissions for 10 August, but GFED

reports the lowest emissions for 22 and 23 August. 10 August is a relatively clear day for the spatial domain, so it is not likely that missing remote sensing data will influence the results due to cloud cover. 22 August has more cloud cover, meaning that the clouds obscure some thermal anomalies. However, the Yosemite Rim Fire area is clear, so it is likely being captured by satellite remote sensing products. More fires are seen from VIIRS throughout the boundary on 22 August than 10 August (Figure 4). The increase in fires is reflected in the inventory emissions results, with all inventories reporting higher emissions and burned area on 22 August than 10 August.

10 August does not correspond to the Yosemite Rim Fire period, while 22 and 23 August do. Additional fire radiative power data from VIIRS shows 22 and 23 August have some of the highest fire activity for the entire year of 2013 (Figures 4 and 5). The differences in fire emissions inventory emissions estimates for 10 August and 22 August provide insight into how the inventories capture this change in fire activity. On 10 August, FINN reported 36.7% of the $PM_{2.5}$ emissions in grams reported on 22 August. GFED reported 66.5% of 22 August $PM_{2.5}$ emissions in grams on 10 August. MFLEI (250 m) reported 70.2% of 22 August $PM_{2.5}$ emissions in grams on 10 August, and MFLEI (10km) reported 31.9% of 22 August $PM_{2.5}$ emissions in grams on 10 August. WFEIS (MODIS) reported 39.9% of 22 August $PM_{2.5}$ emissions on 10 August. VIIRS FRP on 10 August reports 65.8% of the 22 August FRP. This is in line with the percentage from day to day reported by MFLEI (250 m) and GFED. MFLEI (250 m) and GFED are well correlated, as found in this analysis and other literature [15]. Both inventories may be better representing the high emissions days and maximum emissions days than other emissions inventories. This is confirmed by comparing and correlating with VIIRS data, discussed in the previous section. However, other analyses in this paper have found that GFED does not accurately represent the overall temporal distributions of emissions from fires, and no emissions inventory represents the same temporal distribution of fire activity seen in the VIIRS data.

The first frequently detected influential point is 10 August. The VIIRS FRP data for August 2013 shows that 10 August has a higher FRP than the nearby days. VIIRS also showed an increased number of points with FRP on 10 August, corresponding to more fires. Fires are occurring, and the FRP of those fires is higher than the FRP on surrounding days. This single-day increase could be difficult for the Bayesian model to estimate, leading to an increase in importance for this point (i.e., day). 22 August shows significant FRP from VIIRS (Figures 4 and 5). Higher emissions are reported from all emissions inventories on 22 August compared to 10 August or 23 August. 22 August contains the third-highest daily-summed VIIRS FRP for August. This increase suggests high fire activity and thus higher emissions, which is well captured by all fire emissions inventories. 23 August is also a frequent influential point. VIIRS FRP data for 23 August shows a lower FRP than both 10 August and 22 August. The fire emissions inventories that reflect this VIIRS FRP trend are GFED, MFLEI (250 m), and MFLEI (10 km). FINN and WFEIS (MODIS) report that 23 August has lower $PM_{2.5}$ emissions than 22 August but higher emissions than 10 August.

Based on data from VIIRS, 10 August is a lower FRP day than 22 and 23 August. FINN may be capturing the daily variability of fires better than GFED because FINN reports the lowest emissions on 10 August, while GFED reports the lowest emissions on 22 and 23 August. GFED reports higher emissions than other inventories for a lower FRP day and lower emissions on higher FRP days. 22 August is the day with the third-highest FRP reported by VIIRS. High FRP is also reported for 10 August. This could explain why these days are frequently influential in the Bayesian model. Larger emisisons estimates will have more influence on the model.

**Table 4.** Frequent influential points from each model. These dates correspond to the points found most frequently in the influential points analysis. Even on the same individual days that influenced the models, the inventories show vastly different emissions and burned areas each day. The percentage of annual $PM_{2.5}$ emissions emitted on that day is presented for all inventories, and the percentage of annual FRP emitted that day per VIIRS data is also provided for comparison.

| | FINN | GFED | MFLEI (250 m) | MFLEI (10 km) | WFEIS (MODIS) | VIIRS |
|---|---|---|---|---|---|---|
| | 10 | August | 2013 | (3 | occurrences) | |
| $PM_{2.5}$ (grams) | $7.32 \times 10^8$ | $1.25 \times 10^9$ | $1.51 \times 10^{10}$ | $2.51 \times 10^{10}$ | $4.91 \times 10^9$ | – |
| Percent of Annual $PM_{2.5}$ Emissions | 0.62% | 4.16% | 4.14% | 1.36% | 1.86% | 4.31% |
| Burned Area ($m^2$) | $7.02 \times 10^7$ | $4.84 \times 10^8$ | $5.60 \times 10^8$ | $5.60 \times 10^8$ | $3.44 \times 10^8$ | – |
| | 22 | August | 2013 | (8 | occurrences) | |
| $PM_{2.5}$ (grams) | $1.99 \times 10^9$ | $1.88 \times 10^9$ | $2.15 \times 10^{10}$ | $7.87 \times 10^9$ | $1.23 \times 10^{10}$ | – |
| Percent of Annual $PM_{2.5}$ Emissions | 1.66% | 6.29% | 5.88% | 17.07% | 4.66% | 6.55% |
| Burned Area ($m^2$) | $8.93 \times 10^7$ | $1.27 \times 10^8$ | $2.33 \times 10^8$ | $2.33 \times 10^8$ | $2.16 \times 10^8$ | – |
| | 23 | August | 2013 | (4 | occurrences) | |
| $PM_{2.5}$ (grams) | $1.35 \times 10^9$ | $1.05 \times 10^9$ | $1.17 \times 10^{10}$ | $1.13 \times 10^8$ | $1.13 \times 10^{10}$ | – |
| Percent of Annual $PM_{2.5}$ Emissions | 1.13% | 3.51% | 3.22 % | 2.46% | 4.11% | 1.91% |
| Burned Area ($m^2$) | $8.03 \times 10^7$ | $1.03 \times 10^8$ | $1.19 \times 10^8$ | $1.19 \times 10^8$ | $1.15 \times 10^8$ | – |

## 5. Discussion

Although it is impossible to determine which fire emissions inventory best represents real-world conditions, it is possible to investigate the emissions inventories to make an informed choice. Lacking an evaluation data set, selecting any fire emissions inventory becomes difficult, as each has benefits and drawbacks. This section discusses the advantages and disadvantages of each fire emissions inventory based on the results shown above.

### 5.1. Inventory Resolution

The maps of fire locations for each inventory compare each inventory's spatial resolutions and reported fires. The number of fires and their spatial distribution differs, despite many emissions inventories using the same burned area dataset. The table comparing information from the inventories (Table 1) quantifies the difference between the number of fire pixels reported by each fire emissions inventory. For example, for 2013, MFLEI (250 m) has nearly 120,000 different fire pixels, but GFED has 921. Much of this discrepancy is related to the different spatial resolution of each inventory, with MFLEI (250 m) having a much smaller spatial resolution than GFED. The 921 pixels where fires occurred as reported by GFED cover an area of 709,170 km², but the 119,669 pixels reported by MFLEI (250 m) cover 7490 km². This difference in spatial resolution is essential to consider, as it impacts how each fire emissions inventory reports fires.

Each fire emissions inventory has a pixel grid system for reporting emissions that differs from the resolution of the burned area dataset. The burned area dataset is reassigned to the inventories own pixel system, which creates slight spatial discrepancies in the fire pixels reported for each fire emissions inventory. This grid system transformation may play a part in why the MFLEI (250 m) emissions are so high, especially since MFLEI (10 km) emissions are lower than the MFLEI (250 m) emissions. The MFLEI (250 m) grid has a much finer resolution than the MODIS burned area product (250 m × 250 m versus 500 m × 500 m for MODIS burned area), which could lead to resampling in the process of converting to the finer resolution grid. This resampling is then reduced when aggregating to the larger to 10 km product, which may be why the emissions for MFLEI (10 km) are

lower even though both products use the same input data. This is one way the spatial resolution of a fire emissions inventory impacts the estimated emissions.

WFEIS (MTBS) has the highest spatial resolution used to estimate fire perimeters, but it does not provide the daily fire progression data. GFED has a coarse spatial and temporal resolution. The profile used by GFED to distribute emissions to a finer temporal resolution may not accurately represent diurnal cycles of emissions in the western United States. FINN, the MFLEI products, and WFEIS (MODIS) all have a reasonable spatial and temporal resolution, but the MFLEI (10 km) emissions estimates are closer to the other inventories than MFLEI (250 m).

*5.2. Burned Area*

For both the Yosemite Rim Fire and 2013, all inventories report the same magnitude of burned area (i.e., $10^9$ m$^2$ for 2013 and $10^8$ m$^2$ for the Yosemite Rim Fire). Apart from FINN and WFEIS (MTBS), all fire emissions inventories use the MODIS burned area product to determine the burned area input for each emissions inventory. The difference in the reported burned area from each emissions inventory relates to the method used, whether it is a different burned area product or spatial aggregation. All fire emissions inventories have a maximum burned area in August, but there is a significant disparity between the distribution of burned area for the rest of the year. The pixel grid system that each emissions inventory uses to report emissions, which can impact emissions estimates and the spatial distribution of fire location, can also impact burned area estimates.

Notably, there is no burned area difference between the two MFLEI products because of the method used to aggregate burned area and emissions from the 250 m product to the 10 km product. Emissions are provided for the 10 km product already aggregated from the 250 m product, but the burned area is provided as the number of pixels in the 250 m with fuel burned. Since these two burned areas are the same, MFLEI (250 m) assumes that all 62,500 m$^2$ of fuel in a pixel has burned in a fire. This pixel area is still relatively small in remote sensing fires (the MTBS fire detection threshold is nearly 65 times larger), but the assumption that the entire area in a pixel burns can lead to overestimates in the burned area.

FINN uses the MODIS thermal anomalies product to determine burned area, assuming burned area as a function of pixel vegetation cover. The assumed burned area in FINN is highly uncertain. Based on the burned area results of the other inventories, FINN's assumed burned area product based on vegetation cover seemingly underestimates burned area. Based on the spatial distribution maps, the WFEIS (MTBS) burned area product provides better estimates than the MODIS burned area product for this comparison due to the finer spatial resolution. While GFED included a small fire algorithm for determining the emissions contributions of small fires, it has large uncertainties. In addition to the uncertainty, GFED had a smaller burned area for the Yosemite Rim Fire. The small fire product should excel in these situations, which means the coarse spatial resolution of GFED may be hampering the efficacy of the small fire product.

*5.3. Fuels Classification and Emissions Factors*

The number of vegetation categories included in an emissions inventory and the emissions factors used determine how an emissions inventory captures real-world fuel combustion. A fire emissions inventory with fewer vegetation categories may have more emissions uncertainties as more vegetation types are combined in one category. While MFLEI appears to have fewer vegetation categories than the other emissions inventories, they only provide emissions estimates for the contiguous United States. GFED has eight overall vegetation categories, but only three apply to the spatial domain in this paper– boreal forest, savanna, and temperate forest [16]. Most fuel loading for North America in FINN is assigned to tropical forest, boreal forest, and temperate forest, three of the five total vegetation categories included in FINN [17]. The Fuel Characteristic Classification System FCCS used for the fuel bed classifications [37] used in WFEIS [27] provides a more robust

framework for applying emissions factors when combined with the Consume emissions framework. FCCS has numerous specific vegetation categories integrated directly with Consume to calculate emissions. While some emissions inventories may appear to have more categories than others, only WFEIS has a complex vegetation assignment for the domain of interest.

There are a few notable differences in the emissions factors between inventories. MFLEI has an exceptionally high emissions factor for $PM_{2.5}$ in western and northern forests. The emissions factor in MFLEI is nearly double what is used in the other emissions inventories. Determining which emissions factors approach used by a fire emissions inventory is most effective is challenging. MFLEI uses a large emissions factor for $PM_{2.5}$ in forests to better reflect information in new literature. However, MFLEI assigns emissions factors based on three land cover types, a coarse representation of the entire United States. The Consume model used in WFEIS includes emissions factors for flaming and smoldering combustion. This is advantageous because smoldering combustion is a significant contributor to emissions, and many fire emissions inventories do not differentiate between flaming and smoldering combustion when applying emissions factors. It is also worth noting that the emissions factors used in WFEIS are significantly different based on the combustion type (e.g., flaming or smoldering). The differences between emissions factors for the combustion types make it essential for the method WFEIS uses to apportion combustion types properly. GFED provides additional biomes for assigning emissions factors to forests, but the emissions factors used may not represent the most up-to-date research on emissions factors. FINN also assigns emissions factors based on the highly generalized land cover type that may over-aggregate vegetation type. It is difficult to determine which emissions factors are most advantageous to use, as they lack generalizability between fires, making it difficult to determine even a range of accuracy for emissions factors.

### 5.4. Fuel Loading

The influence of fuel loading on the emissions results can be seen in comparing all common emissions and burned areas for 2013. The emissions inventories show the same pattern for emissions (i.e., GFED has the lowest emissions for all species). However, this is different from the distribution for the burned area (i.e., FINN has the lowest burned area, but GFED emissions are lower than FINN). Either fuel loading characteristics or the emissions factors used in the inventories could cause this. Because all emissions inventories do not report information on fuel loading in their products, it is impossible to understand the impact of fuel loading on the emissions estimates definitively.

### 5.5. Influential Points Investigation

Many of these fire emissions inventories have influential points in common, and these points are often the days with the highest percentage of annual $PM_{2.5}$ emissions emitted in a single day. However, each of these emissions inventories provide different magnitudes of $PM_{2.5}$ emissions for the days of influential points. While this influential points investigation helps determine if an emissions inventory represents a similar temporal fire activity compared to VIIRS FRP data, it cannot determine which fire emissions inventory is modeling conditions most accurately. It is beneficial to see under what conditions the temporal fire activity profile from a fire emissions inventory does not match VIIRS FRP data as quality control for specific points. It is also interesting to see which inventories reflect the trends in VIIRS FRP data for their influential points. GFED and MFLEI (250 m) have high correlation coefficients between the emissions and FRP because these inventories best capture some of the fire activity maxima reported by VIIRS.

The variety in the timing of influential points for each inventory during the Yosemite Rim Fire shows that each fire emissions inventory captures a different temporal distribution of emissions for the Rim Fire. The variability in the emissions estimates for the Yosemite Rim Fire is more varied than the variability between emissions inventories for more extended periods, such as several months or a year. When using fire emission inventory estimates

as data inputs for atmospheric dispersion modeling, inventory selection will significantly impact atmospheric dispersion modeling on a per-fire basis versus a longer time scale.

### 5.6. Overall Performance

While GFED has reasonable total emissions of $PM_{2.5}$ for 2013, the coarse spatial resolution is a severe drawback for this emissions inventory. GFED also fails to create a reasonable temporal apportionment of emissions for the Yosemite Rim Fire, showing its limitations on a regional scale. The direct comparison reveals that MFLEI (250 m) is likely overestimating fire emissions, even when compared to MFLEI (10 km). While this could be a product of both the high emissions factor for $PM_{2.5}$ in western forests and the small number of vegetation classifications, since the MFLEI (10 km) product emissions are not as high, this points to the possibility that there may be some resampling error with the smaller spatial resolution product. However, the MFLEI (10 km) product also suffers from unreasonable estimates, showing a very high percentage of annual $PM_{2.5}$ emissions emitted on single days, nearly tripling the highest annual percentage emitted in a day values of the other inventories. WFEIS is advantageous in its complex fuel characteristics and includes both flaming and smoldering combustion emissions calculations. The downsides of the WFEIS products are the higher greenhouse gas emissions than other inventories because the cause of this difference is unknown. The lack of a daily burned area progression for the MTBS product is also a downside. The temporal progression of WFEIS does not closely match the temporal fire activity data from VIIRS, but none of the emissions inventories studied here do. WFEIS (MODIS) does not capture the maxima as well as MFLEI (250 m) or GFED, but the day-to-day fire progression is more reasonable.

### 5.7. Study Implications and Limitations

The nature of modeling biomass burning emissions for fire emissions inventories is highly uncertain. Each emissions inventory takes a different approach to modeling the input variables needed to estimate emissions, creating differences in the final emissions reported by each inventory. Some emissions inventories prioritize the timeliness of data but are forced to assume burned area, which creates significant uncertainties. Other emissions inventories prioritize detecting small fires, but that algorithm also introduces significant uncertainties. Some emissions inventories incorporate data from field or laboratory measurements, but these measurements are also uncertain. Each of these uncertainties is propagated through the fire emissions inventory and into the work that incorporates the estimated emissions (i.e., chemical transport modeling or dispersion modeling).

When selecting a fire emissions inventory for atmospheric modeling, these uncertainties and limitations require a "fit for purpose" approach. Because each inventory has strengths and limitations, the inventory selection should be determined based on the fire or smoke modeling purpose. Our future work will use the fire emissions as inputs to an atmospheric dispersion model to simulate smoke transport and estimate smoke exposures. The smoke exposures will be used in a time-series epidemiologic study to estimate associations between smoke and acute cardiorespiratory health outcomes. In this case, capturing the daily variability in the ambient smoke concentrations is of primary importance. Additionally, we aim to identify smoke contributions from each fire in the domain, requiring a fine spatial resolution inventory. Based on the results in this paper, the inventory selected for our future dispersion modeling is WFEIS, specifically, to combine WFEIS(MTBS) and WFEIS(MODIS). This will provide the best inventory available for our health study, where the MTBS products provide high-quality spatial information about fire location, and MODIS provides the variability of the day-to-day emissions.

While the Bayesian model created for each fire emissions inventory was used for a limited influential points investigation in this paper, there are many ways a Bayesian model can provide additional information. Bayesian measurement error is helpful for this application because the actual value does not need to be known since the true value can be modeled as an unknown variable with a probability distribution [30]. Future studies could

implement a Bayesian measurement error model, where the uncertainties of each variable used in determining fire emissions must be quantified for input into the measurement error model. Further investigation of the inputs into each emissions inventory, such as emissions factors, is required to decrease uncertainties in emissions estimates. Incorporating data from the new generation of geostationary satellites could also improve the inputs to emissions modeling and provide an additional comparison dataset.

## 6. Conclusions

Understanding the type and constituents of wildland fire emissions is crucial to understanding their impacts on human health and climate. A fire emissions model must balance simplifying the real world to be usable and maintaining complexity to represent actual conditions accurately. Each fire emissions inventory uses a different method to model fire behavior, burned area, and emissions. The methodology each fire emissions inventory uses significantly impacts the emissions estimates. Comparing the results from four fire emissions inventories for 2013 revealed how each method of determining emissions impacts the estimates provided by these inventories. The differences between each inventory were exacerbated when investigating a single large fire, showing how inventory selection can vastly impact models that use a fire emissions inventory as an input. Using a direct comparison and Bayesian modeling to understand the differences between fire emissions inventories allows for an informed decision about which fire emissions inventory to use for a particular application.

**Author Contributions:** Conceptualization, S.D.F. and H.A.H.; methodology, S.D.F., H.A.H. and A.G.S.; formal analysis, S.D.F.; investigation, S.D.F., H.A.H., A.G.S. and M.J.S.; resources, H.A.H.; writing—original draft preparation, S.D.F.; writing—review and editing, S.D.F., H.A.H., A.G.S. and M.J.S.; visualization, S.D.F.; supervision, H.A.H. and A.G.S.; project administration, H.A.H.; funding acquisition, M.J.S. All authors have read and agreed to the published version of the manuscript.

**Funding:** This work is supported in part by the National Institutes of Health under award number R01ES029528.

**Institutional Review Board Statement:** Not applicable.

**Informed Consent Statement:** Not applicable.

**Data Availability Statement:** Not applicable.

**Acknowledgments:** This work is supported in part by the National Institutes of Health under award number R01ES029528. We acknowledge the use of imagery and data from the NASA Worldview application (https://worldview.earthdata.nasa.gov/ (last accessed: 19 January 2022)), part of the NASA Earth Observing System Data and Information System (EOSDIS). We acknowledge the use of fire emissions data from FINN ((https://www.acom.ucar.edu/Data/fire/ (last accessed: 19 January 2022)), GFED ((https://globalfiredata.org/pages/data/ (last accessed: 19 January 2022)), MFLEI ((https://www.fs.usda.gov/rds/archive/Catalog/RDS-2017-0039 (last accessed: 19 January 2022)), and WFEIS (https://wfeis.mtri.org/home (last accessed: 19 January 2022)).

**Conflicts of Interest:** The authors declare no conflict of interest. The funders had no role in the design of the study; in the collection, analyses, or interpretation of data; in the writing of the manuscript, or in the decision to publish the results.

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
