# Peer review of "Statistical Comparison and Assessment of Four Fire Emissions Inventories for 2013 and a Large Wildfire in the Western United States"

_fire, doi:10.3390/fire5010027_

Round 1

Reviewer 1 Report

The authors present a very interesting paper about This paper presents a comparison of four emissions inventories and their products: Fire INventory from NCAR (FINN version 1.5), Global 6 Fire Emissions Database (GFED version 4s), Missoula Fire Labs Emissions Inventory (MFLEI (250m) 7 and MFLEI (10km) products), and Wildland Fire Emissions Inventory System (WFEIS (MODIS) and 8 WFEIS (MTBS) products). Results indicate that emissions. The manuscript is clear, relevant for the field and presented in a well-structured manner and scientifically sound. The manuscript’s results are reproducible based on the details given in the methods section. The conclusions are consistent with the evidence and arguments presented. The manuscript is well written and should be of great interest to the readers. However, all figures with charts could be bigger (especially figure 1) and the text too because couldn’t be read. Also, the conclusion should mention more about their future work.  

Reviewer 2 Report

The purpose of this paper is to analyze and compare multiple fire emission inventories. A framework is provided for the lack of a validation dataset, leading to some interesting conclusions, which are instructive for evaluating and utilizing fire emission inventories, but there are still some minor issues that need to be improved. The specific questions are as follows:

  1. Due to the length of this article, it is recommended to add a summary of the article editing in the introduction section
  2. Please correct some mistakes in the article, such as "INventory" on line 63
  3. The article introduces various fire emission inventories and their advantages and disadvantages in 2.1 to 2.4. This is not the result of the article, but there are few relevant references and citations are relatively simple.
  4. In the method part, it is recommended to delete it properly. There are too many text descriptions, which may not actually be needed.
  5. There are a lot of analysis results in the results and discussion part of the article, but in the conclusion of the sixth part, the description is too thin, and the interesting results obtained are not summarized in the conclusion.

Reviewer 3 Report

Wildfires, in the form of bush fires, vegetation fires, forest fires, heath and grass fires, are prevalent throughout the world. Recent high profile events in Australia in 2019 to 2020, the Amazon rainforest in Brazil in 2019 and 2020, the western United States in 2018 and 2020, and British Columbia, Canada in 2017 and 2018, have reminded the global community of the devastating effects uncontrolled fire may cause. There are on average 70 000 forest fires annually in Europe alone. Climate change may increase the risk of increasing wildfire frequency, therefore there is an urgent need to further understand the health effects and public awareness of wildfires. Wood smoke has high levels of particulate matter and toxins. Respiratory morbidity predominates, but cardiovascular, ophthalmic and psychiatric problems can also result. The wider health implications from spreading air, water and land pollution are of concern.

Global-scale fire emissions assessments, by necessity, use generalized or surrogate information to produce emissions estimates. Using a fire emissions inventory is one way to determine the emissions rate and composition of smoke plumes from individual fires. There are multiple fire emissions inventories, and each uses a different method to estimate emissions. The manuscript uploaded by Faustich et al. presents a comparison of four emissions inventories and their products. A combination of techniques, from direct comparison to Bayesian statistical analysis, were used to compare the outputs of each fire emissions inventory.

In the literature on the subject, the authors addressed this problem. In this respect, the work is not original. However, the manuscript submitted for evaluation provides new data.  

Faustich et al., failed to emerge,  which fire emissions inventory best represents real-world conditions. However, they showed, Using a direct comparison and Bayesian modeling to understand the differences between fire emissions inventories allows for an informed decision about which fire emissions inventory to use for a particular application.

In my opinion, the most valuable part of a peer-reviewed manuscript is the discussion. The authors discussed the differences impacting the emissions estimates from the four fire emissions inventories is provided, including a qualitative comparison of the methods and inputs used by each inventory and the associated strengths and limitations.

Comments and suggestions:

  1. Lines 75-193: “Fire Emissions Inventories & Satellite Remote Sensing” - Please describe briefly the theoretical background.
  2. The importance of the presented results in the context of human health should be emphasized more.
  3. Figure 1a-f: page 9: Enlarge the font for the X and Y axes and legends
  4. Reference Lines 841-915: References should be described as follows (Instructions for Authors - Fire), depending on the type of work:

Example - Journal Articles:
Author 1, A.B.; Author 2, C.D. Title of the article. Abbreviated Journal Name YearVolume, page range.
